# Optimal tuning of weighted kNN- and diffusion-based methods for denoising single cell genomics data

**Andreas Tjärnberg**[1,2,3]*, **Omar Mahmood**[4☯], **Christopher A. Jackson**[2,3☯], **Giuseppe-Antonio Saldi**[2], **Kyunghyun Cho**[5,6], **Lionel A. Christiaen**[1,3], **Richard A. Bonneau**[2,3,4,5,6]*

**1** Center for Developmental Genetics, New York University, New York, New York, USA, **2** Center For Genomics and Systems Biology, NYU, New York, New York, USA, **3** Department of Biology, NYU, New York, New York, USA, **4** Center For Data Science, NYU, New York, New York, USA, **5** Flatiron Institute, Center for Computational Biology, Simons Foundation, New York, New York, USA, **6** Courant Institute of Mathematical Sciences, Computer Science Department, New York University, New York, New York, USA

☯ These authors contributed equally to this work.
* andreas.tjarnberg@nyu.edu (AT); rb133@nyu.edu (RAB)

**Data Availability Statement:** All data used in the publication is sourced from publicly available previously published data. Runtime configurations and data for reproducing results are available in the

## Abstract

The analysis of single-cell genomics data presents several statistical challenges, and extensive efforts have been made to produce methods for the analysis of this data that impute missing values, address sampling issues and quantify and correct for noise. In spite of such efforts, no consensus on best practices has been established and all current approaches vary substantially based on the available data and empirical tests. The k-Nearest Neighbor Graph (kNN-G) is often used to infer the identities of, and relationships between, cells and is the basis of many widely used dimensionality-reduction and projection methods. The kNN-G has also been the basis for imputation methods using, *e.g.*, neighbor averaging and graph diffusion. However, due to the lack of an agreed-upon optimal objective function for choosing hyperparameters, these methods tend to oversmooth data, thereby resulting in a loss of information with regard to cell identity and the specific gene-to-gene patterns underlying regulatory mechanisms. In this paper, we investigate the tuning of kNN- and diffusion-based denoising methods with a novel non-stochastic method for optimally preserving biologically relevant informative variance in single-cell data. The framework, *Denoising Expression data with a Weighted Affinity Kernel and Self-Supervision* (DEWÄKSS), uses a self-supervised technique to tune its parameters. We demonstrate that denoising with optimal parameters selected by our objective function (i) is robust to preprocessing methods using data from established benchmarks, (ii) disentangles cellular identity and maintains robust clusters over dimension-reduction methods, (iii) maintains variance along several expression dimensions, unlike previous heuristic-based methods that tend to oversmooth data variance, and (iv) rarely involves diffusion but rather uses a fixed weighted kNN graph for denoising. Together, these findings provide a new understanding of kNN- and diffusion-based denoising methods. Code and example data for DEWÄKSS is available at https://gitlab.com/Xparx/dewakss/-/tree/Tjarnberg2020branch.

manuscript or supplemental material and archived in the OSF database: https://osf.io/cu2br/?view_only=61f47f25d8a54828a65409b7f629fca4.

**Funding:** LC and RB are funded by the NIH R01HD096770 (National Institutes of Health (NIH) https://www.nih.gov/). RB is funded by the NSF IOS-1546218, NSF CBET-1728858 and NIH R01 CA240283 (National Science Foundation (NSF) https://www.nsf.gov/). The funders had no role in study design, data collection and analysis, decision to publish, or preparation of the manuscript.

**Competing interests:** The authors have declared that no competing interests exist.

## Author Summary

Single cell sequencing produces gene expression data which has many individual observations, but each individual cell is noisy and sparsely sampled. Existing denoising and imputation methods are of varying complexity, and it is difficult to determine if an output is optimally denoised. There are often no general criteria by which to choose model hyperparameters and users may need to supply unknown parameters such as noise distributions. Neighbor graphs are common in single cell expression analysis pipelines and are frequently used in denoising applications. Data is averaged within a connected neighborhood of the k-nearest neighbors for each observation to reduce noise. Denoising without a clear objective criteria can result in data with too much averaging and where biological information is lost. Many existing methods lack such an objective criteria and tend to overly smooth data. We have developed and evaluated an objective function that can be reliably minimized for optimally denoising single cell data on a graph, DEWÄKSS. The DEWÄKSS objective function is derived from self supervised learning principles and requires optimization over only a few parameters. DEWÄKSS performs robustly compared to more complex algorithms and state of the art graph denoising methods.

## 1 Introduction

Single-cell RNA-seq (scRNA-seq) experimental methods measure gene expression in individual cells from heterogeneous samples. This allows identification of different cell subpopulations, and has been extensively used to map developmental trajectories. scRNA-seq experiments yield data with hundreds to hundreds of thousands of individual cell observations; however, the measured gene expression in each cell is noisy, due to undersampling caused by the extremely low quantities of RNA present in any individual cell [1]. Many computational applications have been developed that leverage the advantages of scRNA-seq experiments [2–5]. Analysis has primarily focused on the interpretation of the cellular landscape; software suites incorporating customizable workflows have been developed to enable this analysis [3, 6, 7]. Denoising computational approaches to mitigating the sparsity of single-cell data (having few counts per cell) have corrected structural and sampling zeros [8], imputed missing values [9–11], or corrected measured expression values [12–14]. The modeling and motivational assumptions of these approaches vary and include cell-cell similarity, gene covariance, and temporal/trajectory stability.

In any individual cell, some genes will not be detected [15]; genes that have been biologically silenced or repressed and genes that have low stochastic expression may have zero expressed transcripts. Other genes have expressed transcripts in the cell but are measured as zero due to the sampling depth. For some scRNA-seq experimental techniques, there is evidence of zero inflation in measured expression levels [15], but newer droplet-based scRNA-seq methods do not appear to have more zero expression measurements than expected by chance [16]. Single-cell gene expression measurements are a function of transcript sampling depth, which varies widely from technique to technique, stochastic noise in transcript abundance within individual cells, and technical noise which affects genes of all expression levels. Some single-cell denoising methods consider measured zeros to be sampling 'dropouts', and only function to impute non-zero values in place of zero values; these bias corrections to low-expression genes, suppressing variance for these genes by over-correcting zeros and failing to denoise the data in a biologically relevant manner. Other methods are holistic, using the

overall expression profile of all genes to guide denoising, including those which are highly expressed.

The results of these denoising methods for scRNA-seq data vary considerably based on the choice of hyperparameter values, and some methods require selection of an appropriate noise model. Empirical determination of hyperparameter values and noise models may not result in the most optimally denoised results. However, if we define an objective function that measures the performance of our algorithm, we can find an optimal model by choosing hyperparameters that maximize this objective. Self-supervision allows autoencoders to select model parameters to optimally denoise data by using an objective function based on the loss after reconstruction from a lower-dimensional manifold, and these methods have been applied to both genomics in general [12] and single-cell data specifically [9, 17]. However, the self-supervision of an autoencoder does not optimise for the choice of model hyperparameters, such as the loss function or the numbers of units and layers, and there is no principled way to tune these hyperparameters for a given dataset.

One of the most fundamental components of the single cell analysis framework is the k-Nearest Neighbor Graph (kNN-G), which connects each cell to the k cells near it based on the distance between their gene expression profiles. It is used to drive neighbor embedding methods that show the global structure of the data in a low-dimensionality projection [18], to detect communities or clusters of related cells [19, 20], and to establish trajectories through network connections that represent changes over time [21–23]. kNN algorithms are an attractive choice for denoising due to their simplicity, in the simplest case only a single parameter needs to be chosen, but it is still difficult to select optimal model hyperparameters, especially with regard to the use of prior knowledge to tune these algorithms [24]. Diffusion using a kNN-G is the process of averaging over an incrementally larger and larger number of neighbors derived through shared neighbor connections. Looking at a single cell, diffusing using one step means averaging over its neighbors, while a two-step diffusion means averaging over the cells' neighbors *and* their neighbors. One current method for denoising based on kNN-G is MAGIC [14], which diffuses on the kNN-G and maps back to gene expression of single genes. Another denoising approach smooths expression profiles by combining gene expression of kNN-G connected cells [13, 25]. These methods heuristically determine the number of neighbors used, which corresponds to the amount of smoothing applied; this is a drawback when compared to methods that use an objective function that minimizes a desired global objective function.

In this paper we propose using the noise2self self-supervision principle [26]. A method that can be used to constructing an objective function which can be minimized and is self-supervised. With this principle in mind, we constructed an objective function to select kNN-G denoising hyperparameters. This does not depend on an explicit noise model but on an invariant and independent function of the features of the data. We apply this underlying principle to optimally set parameters for denoising single-cell data in a framework called *Denoising Expression data with a Weighted Affinity Kernel and Self-Supervision* (DEWÄKSS), which incorporates a principled self-supervised objective function with weighted kNN-G averaging (Fig 1). We evaluate DEWÄKSS using previously established data and benchmark tests, and compare our self-supervised hyperparameter selection method to the state-of-the-art imputation methods MAGIC [5], DeepImpute [27], DrImpute [11] and SAVER [28]. We find that DEWÄKSS performs at par with or better than other state-of-the-art methods, while providing a self-supervised and hence easily searchable hyper-parameter space, greatly simplifying the application of optimal denoising. We also find that diffusion, although conceptually attractive and previously described as beneficial, is, in fact, not optimal for any method in any setting on any dataset.

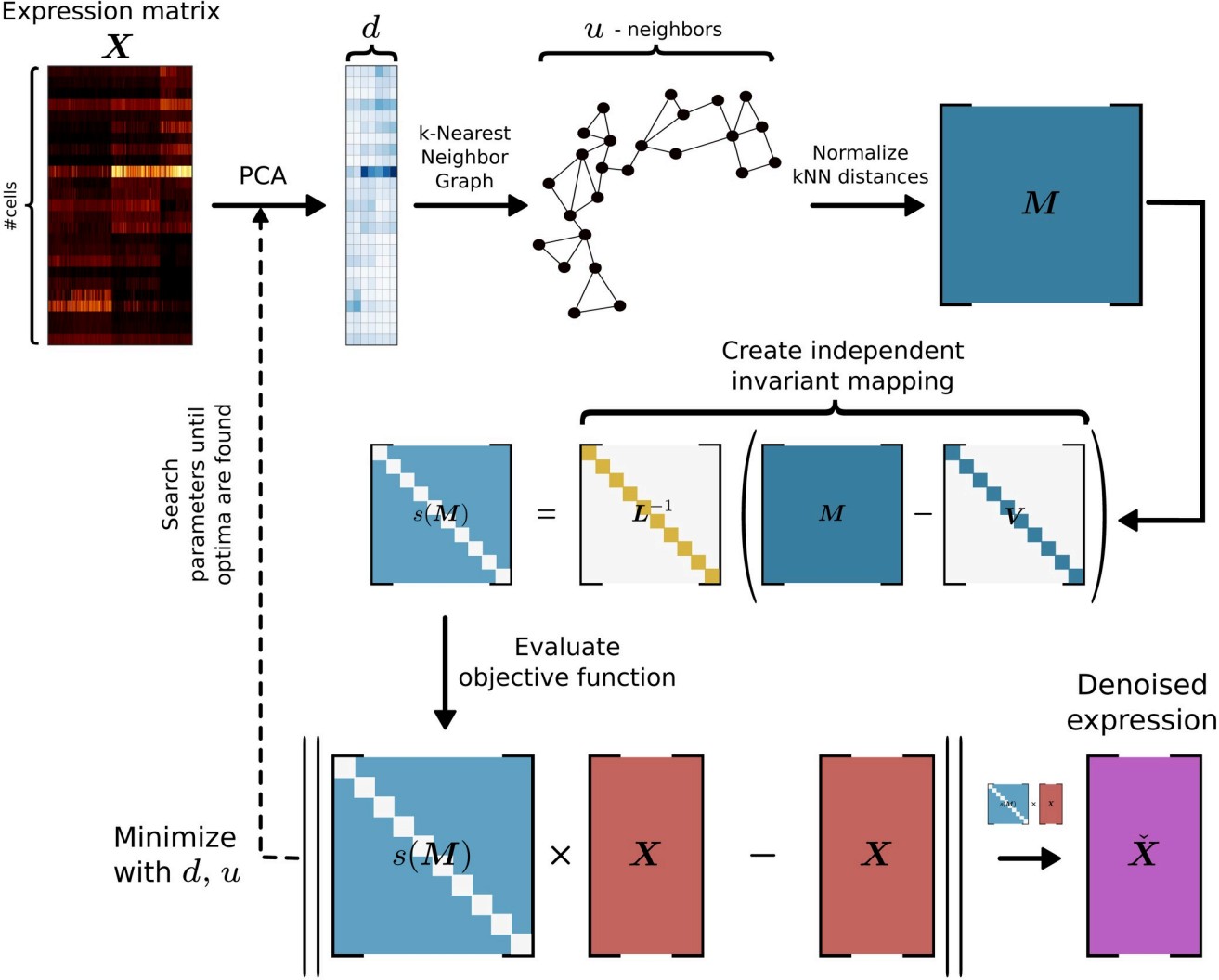

**Fig 1. The denoising process using DEWÄKSS.** The procedure is as follows: (i) The PCA of the expression matrix $X$ is computed. (ii) A cell-to-cell similarity structure is inferred from a k-Nearest Neighbor Graph (kNN-G). (iii) An invariant and independent mapping function is constructed. (iv) The objective function, with a defined optimum for denoising, minimizes the mean squared error (MSE) between the predicted weighted average of the neighbor cells' state and the observed cell state. DEWÄKSS tunes its objective function with two main input parameters, the number of PCs and the number $k$ of neighbors.

## 2 Results

### 2.1 Benchmarking the DEWÄKSS algorithm

DEWÄKSS takes a normalized expression matrix and calculates a smoothed output expression matrix which has denoised gene expression for each cell. The DEWÄKSS expression matrix will have decreased stochastic sampling noise; expression values, including zeros that are likely the result of undersampling, will be weighted according to their sample-to-sample context. We will test the effectiveness of diffusion and kNN-based (diffusion with step size = 1) denoising methods that are tuned with DEWÄKSS objective function using four separate cases.

## 2.2 DEWÄKSS optimally groups cells and performs robustly, independent of data normalization method

Before any denoising can occur, single-cell data generally must be normalized. Recent work has established a benchmark for single-cell analysis methods which is broadly applicable to normalization and denoising techniques [29]. We have therefore compared the performance of DEWÄKSS to several previously-described denoising methods using two benchmark datasets that are generated with different methods. These artificially constructed RNA mixture datasets have a known ground truth; RNAmix_CEL-seq2 is derived from the CEL-Seq2 method [30] and RNAmix_Sort-seq is derived from the SORT-seq method [31]. Within each dataset, 'cells' that belong to the same group are samples that have the same proportions of mRNA from three different cell lines. Any differences between 'cells' in the same group can hence be attributed to technical noise.

Using a computational pipeline [32], we test the effect of normalization methods on denoising techniques, with output scoring defined by a known ground truth (Fig 2). Normalization methods are generally the same as previously tested [29], and include several bulk-RNA normalization methods (TMM, logCPM, DESeq2), several single-cell-specific methods (scone, Linnorm, scran), and a simple Freeman-Tukey transform (FTT). Overall, we find that DEWÄKSS yields expression profiles with high within-group correlation (averaged over all cells in the dataset) independent of the normalization method used, outperforming other denoising methods in the majority of cases (Fig 2A). This is not due to high correlation between cells in different groups (which could indicate oversmoothing), as cells of the same type are strongly correlated and cells of different types are weakly correlated when plotted as a heatmap (S1 Fig).

DEWÄKSS has three essential input hyperparameters: the number of principal components (PCs) for initial data compression, the number of nearest neighbors to build the graph embedding with ($k$), and the connection mode for edge weights (either normalized distances or network density weighted connectivities). For this benchmark, the DEWÄKSS algorithm has grid searched through a model hyperparameter space, testing connection mode {distances, connectivities}, number of neighbors {1, 2, . . ., 20, 30, 40, . . ., 150, 200}, and PCs {1, 2, . . ., 20, 30, 40, . . ., 150, 200}. Self-supervision selects optimal model hyperparameters by minimization of mean squared error (MSE); for the RNAmix_Sort-seq dataset that has been normalized by FTT, this is the normalized distance mode with 4 PCs and 80 neighbors (Fig 2B). The optimal selection of hyperparameters varies from dataset to dataset and by normalization method (Table 1). In most cases, the optimal MSE is found for the parameters given by normalized distances with between 50-130 neighbors and 3-13 PCs, but data which has not been normalized has very different optimal model hyperparameters. In general, using normalized distances as edge weights between neighbors outperforms using connectivities. In all cases, the optimal number of diffusion iterations is 1 (no diffusion) given a specific set of PCs and $k$, indicating that diffusion is not optimal on this data. The small number of PCs and large value for optimal neighbors suggests that this dataset is simplistic with weak local structure, which is a reasonable expectation given the artificial construction of the RNAmix datasets. The MSE is scaled differently depending on the data normalization and should not be compared between methods.

## 2.3 DEWÄKSS maintains cluster homogeneity and deconvolves cluster structure comparably to state-of-the-art methods

To evaluate DEWÄKSS on higher-complexity data, we adapt the benchmark used by DeepImpute [27], using a dataset which contains 33 annotated cell types in primary visual cortex cells

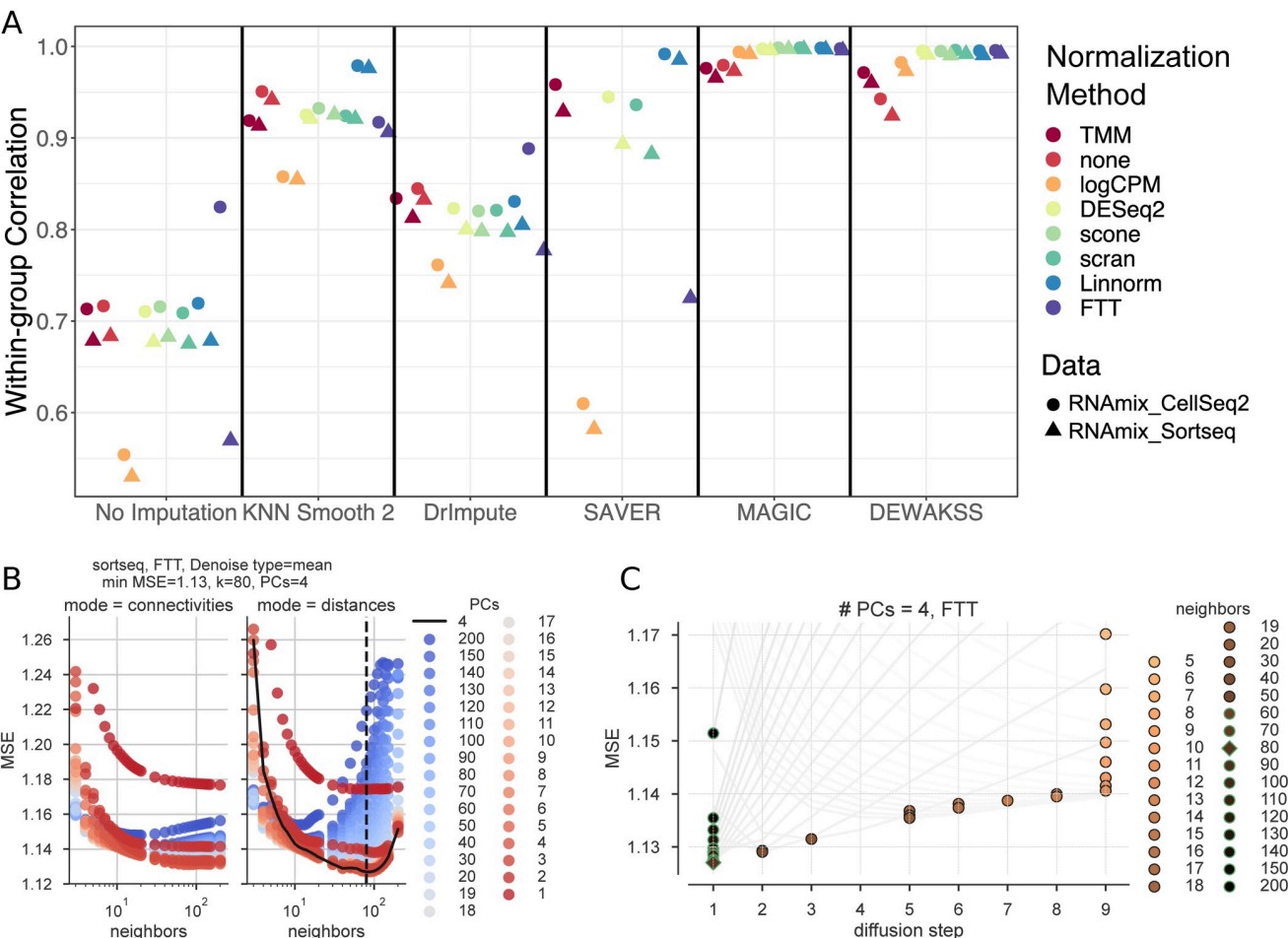

**Fig 2. Benchmarking DEWÄKSS on a predefined benchmark test [29].** A) The average Pearson correlation coefficients between cells that are known to be in the same group, calculated as in Tian et al. [29] for the RNAmix_CEL-seq2 and RNAmix_Sort-seq benchmark datasets. DEWÄKSS yields highly correlated expression profiles for "cells" in the same group, robustly across different normalization methods. B) Self-supervised hyperparameter grid search results (RNAmix_Sort-seq normalized by FTT). Neighbors are on the x-axis and PCs are colored. The optimal configuration neighbors are shown by the dotted black line and PCs are shown by the solid black line. C) Optimization behavior using optimal PCs = 4 found in (B) for 5-200 neighbors. The lowest prediction error for each diffusion trajectory (line) is marked by a circle with a green outline if it corresponds to the number of iterations in the optimal configuration i = 1 or in black when the optimal number of iterations > 1. The optimal value is marked by a diamond. The number of diffusion steps decreases as the number of neighbors increases. The number of diffusion steps is truncated to 9 steps. The prediction error decreases as the number of neighbours increases from 5-80, and then increases.

from mice (GSE102827) [33]. We evaluate performance after preprocessing and denoising the count data by defining experimental clusters using the Leiden algorithm [19], which tunes the number of clusters based on merging and partitioning the kNN-G to find an optimal partitioning given the resolution parameter $r$. Differences between annotated cell types and experimental clusters are quantified by the Fowlkes-Mallows score. The silhouette score estimates the relative within-cluster density of the annotated cell-type clusters compared to the closest annotated neighbor-cluster distances—and is, therefore, independent of the cluster algorithm choice—and evaluates the closeness of known cell identities through different data transformations. The silhouette score is separately calculated on two dimension-reduction projections, (i) using 2 UMAP [18] components, and (ii) using PCA with the number of components used to compute the kNN-G used in the Leiden and UMAP algorithms.

**Table 1. Optimal hyperparameters selected by DEWÄKSS self-supervised objective function.**

| Normalization | Dataset | iteration | mode | neighbors | pcs | MSE |
|---|---|---|---|---|---|---|
| DESeq2 | celseq2 | 1 | distances | 120 | 3 | 0.466 |
| FTT | celseq2 | 1 | distances | 90 | 5 | 0.878 |
| Linnorm | celseq2 | 1 | distances | 110 | 4 | 0.066 |
| logCPM | celseq2 | 1 | distances | 100 | 6 | 4.567 |
| none | celseq2 | 1 | connectivities | 14 | 120 | 4.428 |
| scone | celseq2 | 1 | distances | 120 | 4 | 0.445 |
| scran | celseq2 | 1 | distances | 130 | 3 | 0.484 |
| TMM | celseq2 | 1 | distances | 50 | 6 | 0.378 |
| DESeq2 | sortseq | 1 | distances | 100 | 3 | 0.513 |
| FTT | sortseq | 1 | distances | 80 | 4 | 1.127 |
| Linnorm | sortseq | 1 | distances | 100 | 4 | 0.083 |
| logCPM | sortseq | 1 | distances | 80 | 13 | 4.684 |
| none | sortseq | 1 | distances | 10 | 17 | 5.321 |
| scone | sortseq | 1 | distances | 100 | 4 | 0.484 |
| scran | sortseq | 1 | distances | 120 | 3 | 0.536 |
| TMM | sortseq | 1 | distances | 50 | 6 | 0.412 |

We compare the results of DeepImpute, DEWÄKSS, MAGIC, DrImpute and SAVER to count data that has been preprocessed but has otherwise not been denoised (pp). The preprocessing is detailed in section 4.4, and, in short, consists of filtering and median normalizing the data followed by a Freeman-Tukey transformation. DeepImpute [27] takes as input the raw count data and needs to be preprocessed after, as above. MAGIC is run using the Seurat pipeline [3]. For this dataset, DEWÄKSS selects 100 PCs and 150 neighbors as optimal hyperparameters (Fig 3A)). After denoising we evaluate the performance metrics with a range of

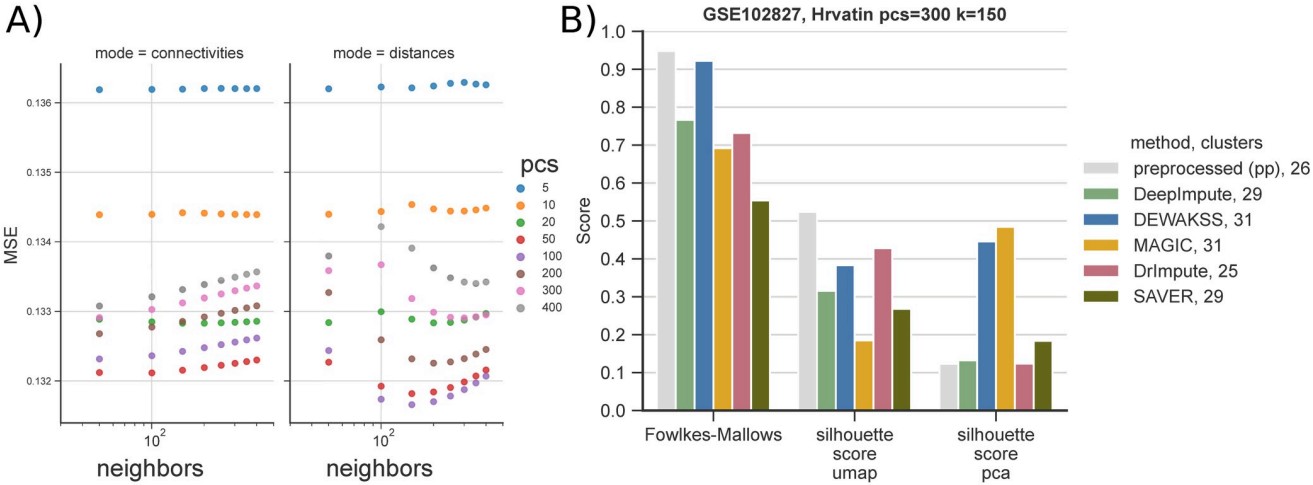

**Fig 3. Denoising celltype-annotated data from Hrvatin et al. [33] using metrics from Arisdakessian et al. [27].** The dataset contains 33 annotated celltypes in 48267 cells. A) Optimal denoising of the expression data with DEWÄKSS requires 100 PCs and $k = 150$ neighbors. B) We benchmark six different denoising pipelines: (i) in-house preprocessed (section 4.4), (pp), (ii) DeepImpute, (iii) DEWÄKSS, (iv) MAGIC, (v) DrImpute and (vi) SAVER. To be able to run (v) and (vi) we down-sample the data to 10% of the annotated cells. After preprocessing & denoising, data is clustered with the Leiden algorithm [19] using 300 PCs and 150 neighbors (resolution is set to $r = 1$ for DEWÄKSS, $r = 2$ for preprocessed (pp) and DeepImpute, $r = 4$ for DrImpute and SAVER, and $r = 0.5$ for MAGIC). Algorithm performance is measured with the Fowlkes-Mallows metric and silhouette score on two representations of the data, PCA and UMAP.

clustering and dimensionality reduction parameters to estimate the sensitivity of the performance metrics (clustering and cell dispersion during projection) to the choices of these parameters (S2 Fig). Overall performance (as determined by MSE) is poor when using few components (PCs) and a small number $k$ of neighbors, which is similar to the default parameters in many processing pipelines (S2(A) Fig). This underlines the importance of carefully considering the amount of variance to be used in the initial kNN-G construction.

Because the number of inferred clusters influences the Fowlkes-Mallows score, we also adjust, by applying a factor, doubling, quadrupling or halving, the resolution parameter $r$ of the Leiden clustering algorithm to increase or decrease the number of clusters to be closer to the number of annotated clusters (33). To be able to run DrImpute and SAVER we down-sample the dataset to 10% of the annotated cells including all 33 clusters before denoising using 4800 cells. $r$ is increased from 1 to 2 for DeepImpute and pp, increased from 1 to 4 for DrImpute and SAVER, and decreased from 1 to 0.5 for MAGIC. DEWÄKSS is not adjusted as the number of clusters falls close to the number of annotated clusters by default.

## 2.4 Optimal kNN denoising does not involve diffusion

On all test datasets, we observed that the optimal configuration was found to have a single iteration (no diffusion) but variable (dataset specific) optimal number of PCs and neighbors (Fig 2C). This observation extended to all normalization methods if parameter spaces with sufficient numbers of neighbors were explored (Table 1). To determine if diffusion is improving denoising for real-world data, we applied DEWÄKSS to seven published single-cell datasets. We tested on mouse bone marrow (BM) data [34], on human cell line epithelial-to-mesenchymal transition (EMT) data [5], on *Saccharomyces cerevisiae* data from rich media (YPD) and on *Saccharomyces cerevisiae* data after treatment with rapamycin (RAPA) [35], on mouse visual cortex tissue (VisualCortex) data [33], on human embryonic forebrain tissue (hgForebrainGlut) data and on mouse dentate gyrus granule neuron (DentateGyrus) data [36]. The BM and EMT datasets are preprocessed following the vignette provided by the MAGIC package [5] (section 4.5). The YPD, RAPA and VisualCortex datasets are preprocessed using the procedure in section 4.4. The hgForebrainGlut and DentateGyrus datasets are preprocessed with the velocyto [36] and SCANPY [4] python packages using the provided vignettes (section 4.6).

We run DEWÄKSS on these datasets (searching for hyperparameters using ∼equidistant values in log space) to find the optimal configuration (S5 Fig). For the BM dataset we let the algorithm run 20 diffusion steps to map out the objective function. For all other datasets we use *run2best*, which finds the first minimum MSE during diffusion and then stops the search (S6 Fig). All six real-world datasets result in optimal MSE when there is no diffusion (number of iterations $i = 1$) (Table 2).

**Table 2. Optimal configurations found by hyperparameter search on DEWÄKSS on seven real-world single-cell datasets.**

| Dataset | iteration | MSE | mode | neighbors | PCs |
|---|---|---|---|---|---|
| BM [34] | 1 | 0.311 | distances | 100 | 50 |
| EMT [5] | 1 | 0.222 | distances | 100 | 100 |
| VisualCortex [33] | 1 | 0.132 | distances | 150 | 100 |
| YPD [35] | 1 | 0.217 | distances | 150 | 50 |
| RAPA [35] | 1 | 0.261 | distances | 175 | 20 |
| hgForebrainGlut [36] | 1 | 0.106 | distances | 100 | 20 |
| DentateGyrus [36] | 1 | 0.055 | distances | 100 | 100 |

## 2.5 DEWÄKSS preserves data variance for downstream analysis

The main goal of denoising scRNA-seq data is to reduce the influence of noise and to reveal biological variance. High dimensional biological data contains some variance which is due to random noise and should be removed, and some variance that is due to biological differences and should be retained. Removing noise is important for correct interpretation of patterns in the data, but attenuating biological variation eliminates biological signal and can result in biased analyses. Biological variance is not easily separable from technical noise, and denoising methods risk oversmoothing, retaining only the strongest patterns (*e.g.* the first few principal components of the data) while discarding informative minor variation. It is therefore critical when tuning model parameters to have an objective function that takes into account the total variance of the data structure.

We evaluate the effect that denoising has on data variance by comparing the singular value structure of the denoised data for different methods, which represents the relative variance of all dimensions. Although the optimal amount of variance and the number of components that capture that variance is not known, we reason that comparing the relative variance in all dimensions allow us to determine the extent to which a denoising method is smoothing the data. Denoised data that requires fewer components to capture most (>90%) variance is more smoothed. When most variance is compressed into a handful of principal components, features within the data become collinear, more complex interactions between features disappear, and only the strongest sources of variance are preserved. Some downstream analyses are likely to be more sensitive to this oversmoothing than other analyses. For example, a clustering approach may still effectively separate groups based on variance in only a few dimensions, but regulatory inference may be substantially confounded or uninterpretable.

MAGIC [5] is currently among the most popular algorithms for denoising single-cell RNA-seq data. It uses a heuristic for determining optimal smoothing; as published, it used $\Delta R^2$ between diffusion steps, but the most recent implementation has switched to Procrustes analysis of the differences between diffusion steps. Neither approach has an objective way to determine optimal smoothing. In the absence of crossvalidation or some other external method that prevents overfitting, we expect $R^2$ to decrease until all data is averaged, *i.e.*, to saturation, and a Procrustes analysis should behave similarly. MAGIC addresses this by hard-coding a stopping threshold which determines when the data is smoothed "enough"; because this threshold is not data-dependent, it can result in highly distorted outputs [27, 35, 37].

If this threshold is converted to a model hyperparameter, it is still necessary to tune with some external method as it has no lower bound for arbitrarily poor estimates.

We compare the effects of denoising using a heuristic as implemented in MAGIC [5], using DEWÄKSS in its optimal configuration and using DEWÄKSS in an oversmoothing (non-optimal) configuration for comparison. We also run this comparison for DeepImpute [27], DrImpute [11] and SAVER [28] with default configurations. This comparison is performed on the previously-described mouse BM, preprocessed using the approach described in [5]. We run MAGIC with three sets of model parameters; the default parameters, default with early stopping (the diffusion parameter $t = 1$), and with the decay parameter $d = 30$. For DEWÄKSS, we scan a log-equidistant parameter range for the optimal configuration (S5(A) and S6(A) Figs) and find that the optimal configuration uses normalized distances with number of neighbors $k = 100$ and with number of principal components PCs = 50 for $i = 1$, giving MSE = 0.3107. Diffusion iterations $i$ increment until a minimum MSE is found. For the BM data, the MSE generally decreases as the number of PCs increases. Beyond a certain point, however, continuing to increase the number of PCs (to 500) increases the MSE. The optimal number of PCs, 50, is small compared to the size of the data and suggests that some compression of the data is

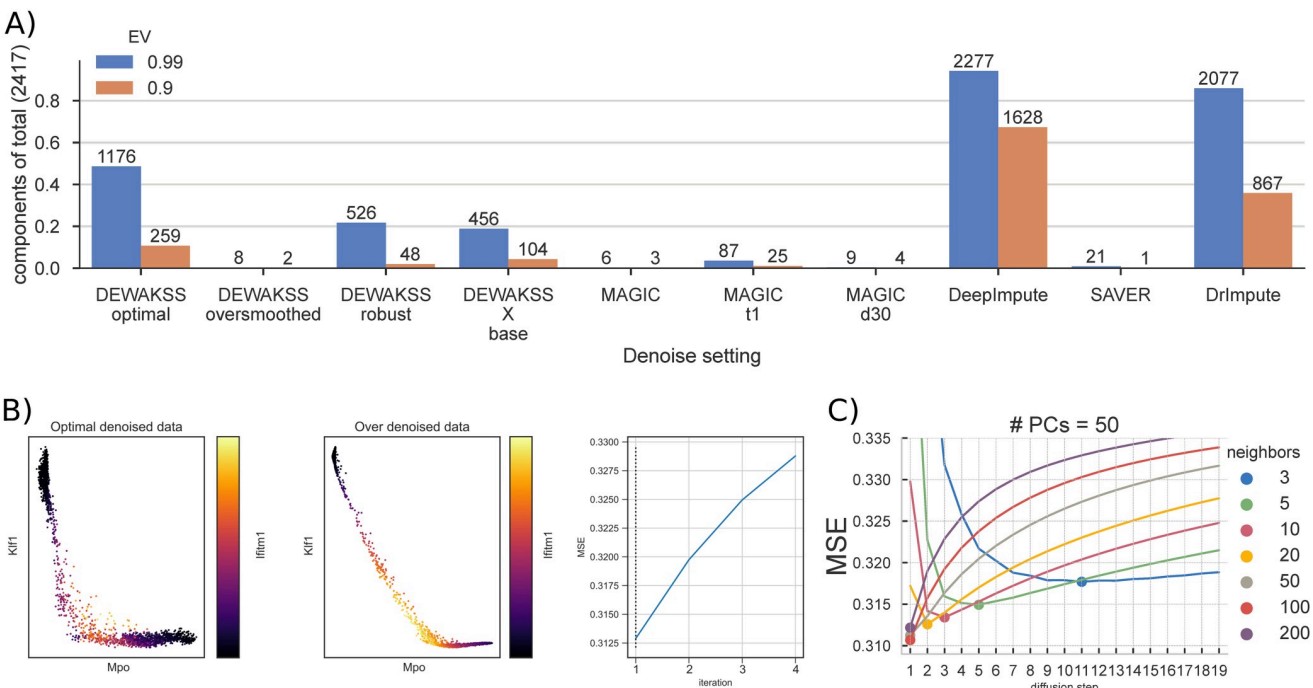

**Fig 4. Mouse bone marrow (BM) data denoising.** A) The numbers of principal components needed to explain 99% and 90% of the variance in the data for different hyperparameter values for DEWÄKSS and MAGIC. DEWÄKSS is run with **optimal** parameters ($k = 100$, $PCs = 50$, $i = i_{\mathrm{minMSE}}$), with **oversmoothed** parameters ($k = 100$, $PCs = 50$, $i = i_{\mathrm{minMSE}}$), with **robust** parameters ($k = 10$, $PCs = 13$ selected using MCV as in S1 Section, $i = i_{\mathrm{minMSE}}$), and as **X base**, where normalized expression values are used instead of PCs with ($k = 100$, $i = i_{\mathrm{minMSE}}$). MAGIC is run with **defaults** ($d = 15$, $PCs = 100$, $k = 15$), with early stopping **t1** ($t = 1$), and with **d30** ($d = 30$). B) Expression of erythroid marker gene Klf1, myeloid marker Mpo, and stem cell marker Ifitm1 in DEWÄKSS optimal and DEWÄKSS oversmoothed data. The MSE increases in each iteration. C) The objective function output as a function of diffusion steps for the optimal number of PCs = 50. The minimum MSE is found for 100 neighbors and 1 diffusion step, *i.e.*, using only the selected 100 neighbors.

optimal before running the kNN algorithm. To oversmooth the data we extended the number of iterations to run DEWÄKSS to $i = 4$, beyond the optimal number of iterations (Fig 4B). We also used the molecular cross-validation (MCV) application of noise2self [38], implemented as in S1 Section, to select the optimal number of PCs for denoising by DEWÄKSS. We found that MCV selected fewer PCs for denoising compared to the DEWÄKSS objective function (S8 Fig; 13 PCs compared to 50 PCs).

To investigate how the variance structure of the data changes based on denoising we compute the singular values (section 4.7) and determine the number of components needed to explain 90% and 99% of the variance for each dataset after denoising (Fig 4A and S3 Fig). We observe a striking difference between the oversmoothed data and the optimally denoised data. With optimal denoising, 90% of the variance is captured by 259 components. Utilizing the MCV method to select hyperparameters resulted in an intermediate amount of variance retained after denoising when compared to the DEWÄKSS optimal and the DEWÄKSS oversmoothed denoised data. Only 2 components are needed to capture 90% of the post-processing variance when oversmoothing the data, showing that a substantial portion of the original information content of the data is lost in this regime. DEWÄKSS oversmoothing is comparable to the results of using MAGIC with default parameters, where 90% of the variance in the post processed data can be captured with only 3 components. When using only one iterative step and default parameters, MAGIC captures this amount of variance using 25 components. In most cases the MAGIC algorithm generates shallow variance structures with a few components needed to express nearly all of the variance. The variance structure can differ greatly

depending on the hyperparameters chosen for DEWÄKSS, and poor parameter selection results in shallow variance structures. However, the objective function automatically identifies an optimal configuration such that we expect to keep the relevant variance. For SAVER the first component capture almost all the variance while the 21 subsequent components capture the remaining variance implying that SAVER compress the data significantly. For DeepImpute and DrImpute the opposite is the case that the 99% of the variance are captured by over 80% of the components implying that these methods maintains a more flat distribution of the data over the components.

We can see the consequence of oversmoothing when plotting the expression of the erythroid marker Klf1, the myeloid marker Mpo, and the stem cell marker Ifitm1 (Fig 4C). Very few individual cells express both Klf1 and Mpo in the optimally-denoised data, but the oversmoothed data implies that there is a smooth continuous transition from high Klf1 expression, through co-expression of Klf1, Mpo, and Ifitm1 markers, to high Mpo expression. Although the difference in MSE is not large ($\Delta$ MSE < 0.0175) between these two denoised datasets, the resulting biological interpretation differs a great deal, and likely highlights a spurious relationship in the oversmoothed case.

We run a similar analysis on the EMT data comparing DEWÄKSS and MAGIC (S4 Fig) and find identical effects.

### 2.6 DEWÄKSS improves recovery of differentially expressed genes

Biological analysis of gene expression data often requires determining differentially expressed genes (DEGs) between groups of cells. We evaluate the effect of denoising on discovery of DEGs by comparing DEGs from a subset of cells with genetic perturbations [35] to the equivalent bulk microarray data [39]. Eleven different gene deletion strains are compared to a wild-type control. Five of these gene deletion strains have *few* DEGs versus six strains with *lots* of DEGs; gene deletion strains with more than 63 DEGs (1% of genes in the yeast genome) in the bulk data are considered to have lots of DEGs. DEGs from the single-cell data are determined by wilcoxon rank sum test with Benjamini/Hochberg correction $\alpha$ = 0.01.

Most methods increase DEG recovery from single-cell data compared to preprocessing alone (Fig 5), with SAVER performing the best and DEWÄKSS in second. However, performance on test subsets with few DEGs is generally low for all methods (S9 Fig), although these results may be less reliable due to the large effect of single DEGs on performance metrics.

### 2.7 DEWÄKSS scales to large single-cell data sets

Single-cell data sets are continuing to grow in scale, and therefore denoising algorithm performance is an important consideration. We have benchmarked several denoising methods on a standard laptop (details in 4.9) in order to evaluate speed and scalability. The size of the data-sets measured are detailed in S1 Table. We find that in our desktop-scale test, DEWÄKSS is able to analyze the largest data set (64.8k cells x 18.1k genes) in a reasonable time (Fig 6), although DeepImpute and MAGIC are faster. Other methods are not able to run to completion in the larger data sets with the computational resources provided for this test.

## 3 Discussion

In this paper we have introduced a novel objective function, based on noise2self [26], and applied it to self-supervised parameter tuning of weighted k-nearest neighbors (kNN) and diffusion-based denoising. The resulting algorithm, DEWÄKSS, is specifically designed to denoise single-cell expression data. The objective function has a global objective that can be minimized, removing the need to use often unreliable heuristics to select model parameters,

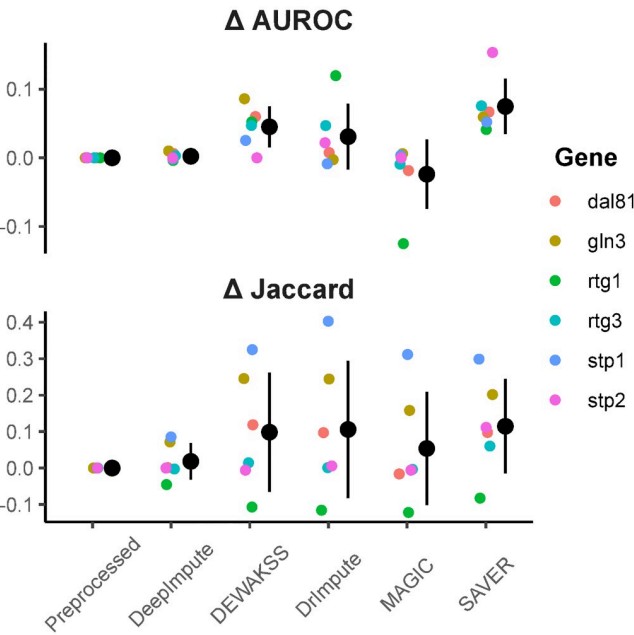

**Fig 5. Differentially expressed genes (DEGs) between bulk and single cell data.** Top panel is the delta AUROC for single-cell DEGs ordered by adjusted p-value for each separate deletion strain. Bottom panel is the delta Jaccard index between bulk DEGs and single cell DEGs at FDR = 0.01. *Preprocessed* is count normalized and log-transformed with no denoising method. Delta is taken between denoised and preprocessed. All computed metrics can be found in S2 Table.

which is a drawback to many current single-cell denoising methods. We demonstrate that this framework accurately denoises data by benchmarking against previously established methods, and find that it is robust to choice of normalisation method (Section 2.2).

Due to the difficulty in establishing a ground truth for most real-world single-cell data, denoising algorithms are frequently tested on synthetic or artificial data. Maintaining biological variance is a crucial aspect of denoising; common downstream applications such as cell type identification, differential gene expression, marker identification, and regulatory network inference rely on biological variance to function. We therefore believe that it is necessary to extensively test on experimental data (a notable strength of Dijk et al. [5] is testing on real-world data). On larger datasets with higher complexity, DEWÄKSS performs well in terms of deconvolving cell types. We find that in general, the amount of variance included when clustering the data has a large impact on the performance of all methods tested, and that DEWÄKSS outperforms other denoising algorithms in this area. While it is still an open question how much variance should be used to project and cluster single-cell data, it is clear that it is an essential component of accurate interpretation.

To investigate the properties of our method we run the algorithm on seven different published single-cell gene expression datasets. In all cases, the optimal denoising configuration (as determined by the objective function) uses the closest neighborhood, and is not improved by diffusion on the kNN graph. Diffusion causes a decrease in denoising performance, compressing almost all of the variance into a handful of dimensions. This may have some advantages for visualizing high-dimensional gene expression data, but most non-visualization analyses are impaired by the loss of variance. We also find that the number of neighbors $k$ and the number of principal components to use tend to be large compared to the default parameters of other methods and conventions used in computational pipelines. In general, there is an advantage to the inclusion

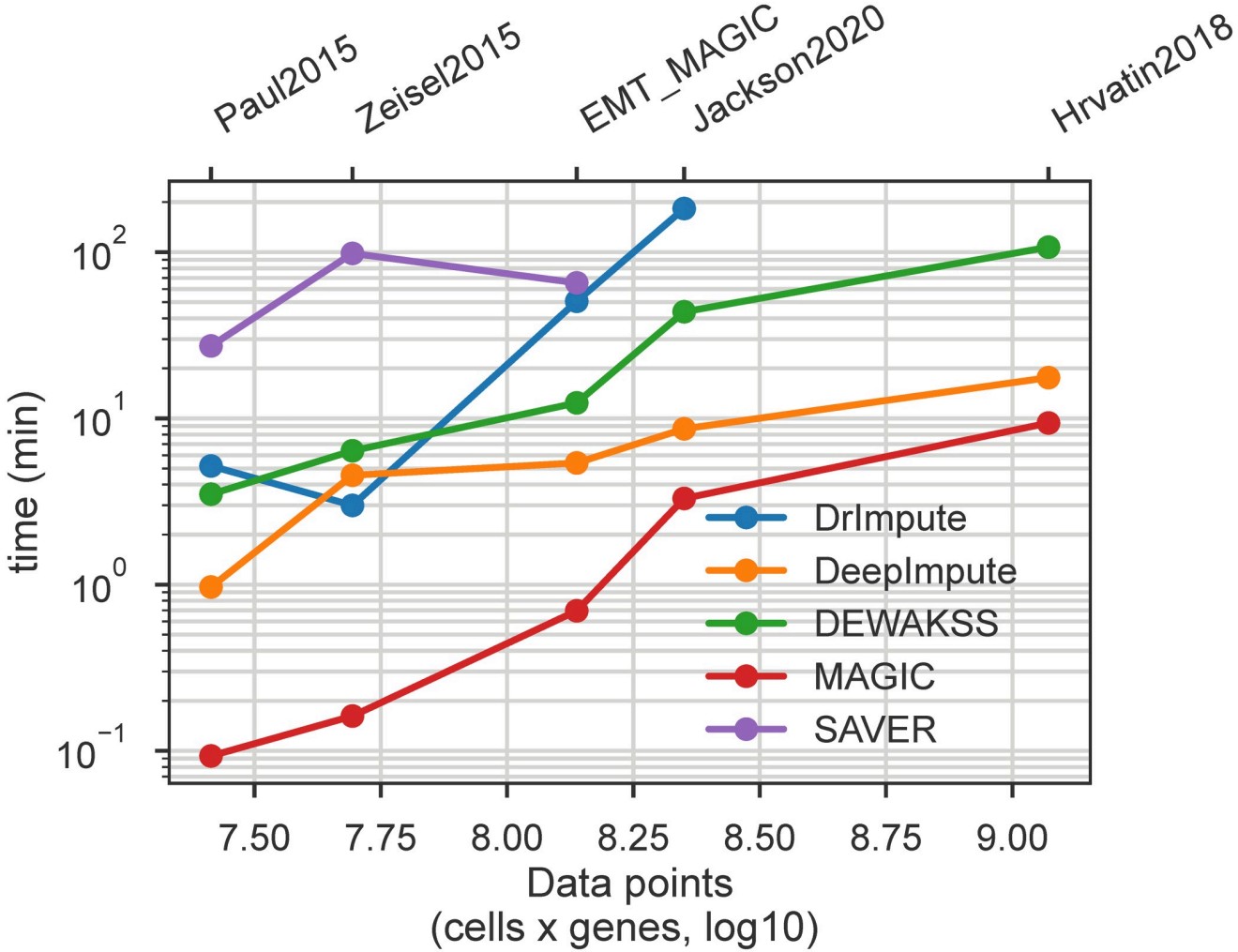

**Fig 6. Computational performance of all tested method on selected datasets.** Runtime (minutes) is plotted against the total number of values (cells *
genes) in the dataset, to account for differing numbers of genes in each data set. Complete results table is available in S1 Table.

of more principal components than called for by common rules of thumb, like using the knee in an explained variance vs number of components plot. However, including an arbitrary number of principal components is not ideal, as excess principal components do decrease performance. Comparing the use of a distance matrix versus the use of a connectivity matrix as a representation of the kNN-G shows that a distance matrix yields better results. The degree of similarity between the expression profile of one cell to that of another cell is relevant for denoising, not just whether cells are more or less similar than other cells' expression profiles in the experiment.

Overall, the DEWÄKSS framework presented here has substantial advantages over heuristic parameter selection. Heuristic-based denoising methods set hyperparameters without a clear basis for effectiveness, often with opaque reasoning for choices. At best, this is likely to result in sub-optimal denoising performance; at worst, it may result in data that is dominated by oversmoothing effects, and which yields incorrect biological interpretations. Our objective function-based method provides a rigorous way of choosing an effective configuration. The difficulties of evaluating how to denoise single-cell data should not be underestimated. It is vital that the effectiveness of single-cell processing methods be quantifiable, so that the

methods can be tuned for performance. We have chosen to use Euclidean distances for all analysis, but DEWÄKSS can accept any graph derived with any distance metric to create the kNN matrix. By constructing a denoising function that uses a $k$-nearest neighbors graph and is consistent with the conditions laid out in noise2self, we have derived an easily-evaluated method that can denoise single-cell data in a self-supervised manner. The DEWÄKSS objective may also have applications to other graph-based algorithms.

## 4 Methods

We begin by presenting a review of the mathematical constraints on our denoising function. We then present the core DEWÄKSS method and objective function. We end with descriptions of our preparatory and preprocessing methods.

### 4.1 Fundamental principle of noise2self

Batson and Royer [26] present the approach noise2self and applied it for UMI counts as the method Molecular Cross Validation (MCV) [38], in which they partition observed features (in this case raw transcript counts) $x_{i \in \mathbb{J}}$, $\mathbb{J} = \{1, \ldots, 2m\}$, into two groups $\{X_J, X_{J^c}\} = \{\{x_1, \ldots, x_m\}, \{x_{m+1}, \ldots, x_{2m}\}\}$ where the superscript $c$ represents the complement set. The task is then to find an invariant function $g(x)_J : \mathbb{R}^{2m} \to \mathbb{R}^{2m}$ that operates only on $X_{J^c}$ and yields an output $\hat{x}$ whose entries at indices $J$ are predictors of $x_J$. This function is independent of $x_J$; some of the features of each datapoint are predicted using another independent set of features of the same datapoint. MCV was implemented for reference (S1 Section) and for selecting a 'robust' set of PCs in section 2.5. MCV applied for PCA component selection optimally selects linearly separable components that are informative and the data is constrained to that sub-selection. S8 Fig. shows the recreated analysis done by Batson and Royer [26]. This practical implementation of the original noise2self principle does not employ a kNN graph and is linear as oppose to the non-linearity of the graph appraoch.

### 4.2 Denoising expression data with a weighted affinity kernel and self-supervision

In DEWÄKSS, as in other state-of-the-art methods, we start by computing a lower, $d$-dimensional representation of the data using PCA. We then compute a connectivity or distance-weighted kNN-G with $u$ neighbors. Our approach is similar to that of MAGIC [5] but differs in two key ways: (i) we use a self-supervised objective function for hyperparameter selection, and (ii) we denoise on the expression values directly to avoid a loss of information/variance that results from overreduction of dimensionality (reducing the data to a latent representation with low rank or low dimensionality such that key biological variation is lost). To calculate the kNN-G, we use the algorithm UMAP [18] and its implementation [40] and create a right-stochastic matrix $M_{d,u}$ from the connectivity/distance matrix. In practice, any graph can be provided as input to DEWÄKSS for denoising. We use the UMAP neighbor algorithm due to its versatility and speed, but alternative methods could be used here. The UMAP implementation only uses the neighbour search algorithm if the number of cells is above 4096 by default and otherwise computes all distances and picks the k closest ones.

Denoising using a normalized kNN-G $M$ can be carried out through simple matrix multiplication

$$\begin{aligned} \check{X} &= MX \\ \check{X}_2 &= M\check{X} \end{aligned} \tag{1}$$

and so on, where $X$ is the expression matrix for $z$ cells, with each column $x_{*j}$ containing the expression values of a single gene $j$ for each cell $k \in K = \{1, \ldots, z\}$, $\check{X}$ is $X$ after one step of denoising and $\check{X}_2$ is $X$ after two diffusion steps of denoising. For a given gene $j$ in a single cell $k$, this equation calculates the weighted average influence of the expression value of the same gene in each neighboring cell. The expression value $\check{x}_{kj}$ is hence set to this weighted sum:

$$\check{x}_{kj} = \sum_{\hat{k}=1}^{z} \mu_{k\hat{k}} x_{\hat{k}j} \tag{2}$$

where $\mu_{k\hat{k}}$ is the $k\hat{k}$-th element of $M$ and $\sum_{\hat{k}=1}^{z} \mu_{k\hat{k}} = 1$.

In general, a Markov process can be forward-iterated as

$$M^2 = MM \tag{3}$$

for a two-step iteration, generalized to $M^n$ for an $n$-step forward process. Denoising is then carried out as follows:

$$\check{X}_n = M^n X \tag{4}$$

In DEWÄKSS we implement a self-supervised procedure by noting that the operation in Eq 4 is the application of an invariant function $g(x)_J$ on each entry of $X$ if the diagonal elements of $M$ at each step $n$ are set to 0 (to enforce the independence noted in section 4.1). If diagonal elements are not forced to zero the second step of the diffusion becomes self referential and enforces overfitting of the objective function (S7 Fig). Here $J = \{j\}$, so the expression value of a gene $j$ in cell $k$ is calculated using the expression values of $j$ in all cells except cell $k$. Eq 2 reduces to:

$$\check{x}_{kj} = \sum_{\hat{k} \neq k}^{z} \mu_{k\hat{k}} x_{\hat{k}j}. \tag{5}$$

That is, for each gene there is an invariant function over the expression values of the gene across all cells.

The Markov property guarantees the independence of the neighborhood graph from past steps. At each step, we set diag $M = 0$ and renormalize to a right stochastic matrix. Let $s$ be a function that removes the diagonal elements of a matrix and then normalizes the resulting matrix to a right stochastic matrix. Let $\mu_{k\hat{k}}^{(d,u,n)}$ be the $k\hat{k}$-th element of $M_{d,u}^{\tilde{n}}$. Then

$$s(M_{d,u}^{\tilde{n}}) = L_{d,u,n}^{-1}(M_{d,u}^{\tilde{n}} - V_{d,u,n}) \tag{6}$$

for each $n$ with

$$V_{d,u,n} = \begin{bmatrix} \mu_{1,1}^{(d,u,n)} & & \\ & \ddots & \\ & & \mu_{z,z}^{(d,u,n)} \end{bmatrix} \tag{7}$$

and

$$\boldsymbol{L}_{d,u,n} = \begin{bmatrix} \sum_{\hat{k} \neq 1} \mu_{1,\hat{k}}^{(d,u,n)} & & \\ & \ddots & \\ & & \sum_{\hat{k} \neq z} \mu_{z,\hat{k}}^{(d,u,n)} \end{bmatrix} \tag{8}$$

with the following notation:

$$\begin{aligned} \boldsymbol{M}_{d,u}^{\bar{2}} &= s(\boldsymbol{M}_{d,u})s(\boldsymbol{M}_{d,u}) \\ \boldsymbol{M}_{d,u}^{\bar{3}} &= s(\boldsymbol{M}_{d,u}^{\bar{2}})s(\boldsymbol{M}_{d,u}). \end{aligned} \tag{9}$$

Eq 4 can be rewritten and incorporated into the mean square error minimization objective as follows:

$$d^*, u^*, n^* = \arg\min_{d,u,n} \|s(\boldsymbol{M}_{d,u}^{\bar{n}})\boldsymbol{X} - \boldsymbol{X}\| \tag{10}$$

$$\boldsymbol{X}^* = s(\boldsymbol{M}_{d^*,u^*}^{\bar{n^*}})\boldsymbol{X} \tag{11}$$

We use this equation to find $n$ that denoises $\boldsymbol{X}^*$ enough to best capture its underlying structure while attenuating variation due to noise. To make sure we keep the invariant sets independent we consider each step an independent Markov process and apply the function $s(.)$ at each step.

## 4.3 Converting kNN to a right stochastic transition matrix

To allow for the use of UMAP distance metrics, DEWÄKSS uses the transformation on distances used in [5],

$$\boldsymbol{M} = e^{-(\boldsymbol{D}/\bar{d}).^{\alpha}} \tag{12}$$

where $\boldsymbol{D}$ is the matrix of distances between the gene expression profiles of different cells and $\bar{d}$ is the mean of the nonzero elements of $\boldsymbol{D}$. A decay rate $\alpha$ is also used, and the . in the equation indicates element-wise operation. This decay rate is applied on a connectivity matrix as $\boldsymbol{C}^{\alpha}$, where $\boldsymbol{C}$ has elements $c \in [0, 1]$. It should be noted that in [5], the decay rate is also applied during the construction of the kNN-G to estimate distance thresholds and there may not be a 1-to-1 correspondence in the algorithms. To stabilize the denoising procedure, the final step before normalizing to a right stochastic transition matrix is to symmetrize $\boldsymbol{M}$ so that

$$\boldsymbol{M} = \frac{\boldsymbol{M} + \boldsymbol{M}'}{2}$$

## 4.4 Preprocessing of scRNA-Seq expression data

We preprocess single cell RNA-seq datasets before applying denoising. Unless otherwise stated, all datasets are preprocessed using the same steps if no guidelines were provided from the dataset or publication source documents.

Preprocessing is carried out using the following steps: (i) Filtering cells by using only those that have greater than a certain number of expressed genes and greater than a certain total cell UMI count. This is done mainly by visual inspection, clipping $1 - 2\%$ of the data—in practice, removing the top and bottom 0.5 percentile of data points; (ii) Removing genes not expressed

in more than $n$ cells, with $n \leq 10$; (iii) Normalizing the counts of each cell so that each cell has the same count value as the median cell count over the cell population; and (iv) applying the Freeman-Tukey transform (FTT) [41] with an adjustment term −1 to preserve sparsity,

$$\tilde{\text{FTT}}(x) = \sqrt{x} + \sqrt{x+1} - 1 \tag{13}$$

The FTT stabilizes the variance of Poisson distributed data. Wagner et al. [13] showed that the FTT is a good choice for single-cell data compared to the log-TPM and log-FPKM transforms as it does not underestimate the relative variance of highly expressed genes and thus balances the influence of lowly expressed gene variation. In other words, the relative variance of highly expressed genes versus more lowly expressed genes should be preserved after transformation. This is an essential property for inferring a relevant kNN-G.

We process all data with the help of the SCANPY framework [4]. DEWÄKSS can accept the SCANPY AnnData object, a regular numpy array or a scipy sparse array as input.

## 4.5 Preprocessing for comparison with MAGIC

The BM dataset is preprocessed using the same approach as used by [5], as detailed here: https://nbviewer.jupyter.org/github/KrishnaswamyLab/MAGIC/blob/master/python/ tutorial_notebooks/bonemarrow_tutorial.ipynb) The EMT dataset is preprocessed as detailed here: https://nbviewer.jupyter.org/github/KrishnaswamyLab/magic/blob/master/python/ tutorial_notebooks/emt_tutorial.ipynb).

## 4.6 Preprocessing hgForebrainGlut and DentateGyrus data

The hgForebrainGlut and DentateGyrus datasets are preprocessed by replicating the process provided by velocyto [36] here https://github.com/velocyto-team/velocyto-notebooks/blob/ master/python/hgForebrainGlutamatergic.ipynb and here https://github.com/velocyto-team/ velocyto-notebooks/blob/master/python/DentateGyrus.ipynb. The package SCANPY is used to carry out the computations [4].

## 4.7 Preservation of variance and PCA computation

To estimate the variance structure of our expression matrix before and after denoising, we take the standard normalization of each variable $j$ in the data so that each observation $k$ in column $j$ is

$$\check{x}_{k,j} = \frac{x_{k,j} - \mathbb{E}_k \, x_{k,j}}{\sigma(x_j)} \tag{14}$$

with $\mathbb{E}_k$ denoting expected value with respect to $k$ and $\sigma$ indicating standard deviation.

Computing the singular value decomposition, we get

$$\check{X} = USV^T \tag{15}$$

The singular values are invariant to the transpose, meaning that they explain the variance in the data independent of whether the data is projected onto cells or genes, and nonzero singular values are bounded by min $\{m, n\}$. To estimate the rank of $\check{X}$ and the nonzero singular values we use the cutoff from numpy [42]:

$$S \leq \max(S) \times \max\{m, n\} \times \epsilon \tag{16}$$

with $\epsilon$ being the machine precision of a numpy float 32 type.

The relative variance is then calculated as

$$\eta_i^2 = \frac{s_i^2}{\sum_i s_i^2} \tag{17}$$

The relative condition number of each singular value can be calculated as

$$|\kappa_i| = \frac{s_i}{\underline{s}} \tag{18}$$

with $\underline{s}$ representing the minimum nonzero singular value defined by Eq 16.

### 4.8 Processing Tian et al. [29] benchmark data

In order to evaluate DEWÄKSS and existing methods in accordance with the benchmark analysis of Tian et al. [29], we use the R code provided in Tian et al. [43] to apply all normalization methods that we can successfully run on the RNAmix_CEL-seq2 and RNAmix_Sort-seq datasets. We also apply our FTT-based preprocessing method on the data in Python and combine the result with the other normalization results into a single data structure.

We then use the same codebase to run the denoising (imputation) methods in [29] on the output of each of the normalization methods on each dataset. We transfer each of these outputs to Python, perform a hyperparameter search using DEWÄKSS on it and record the best parameter configurations along with the corresponding mean squared error (MSE) in Table 1. We apply DEWÄKSS on each normalized input using the optimal configuration for that input and transfer the results back to R, combining them with the other denoising results into a single data structure. Note that some normalization-imputation method combinations are not represented in our figures as we could not successfully run these using the provided pipeline. We use the postprocessing and plotting scripts in [43] to generate plots for our analysis.

### 4.9 Benchmarking computational performance

To benchmark the computational cost of DEWÄKSS we select data sets with varying number of genes and cells from the previously used datasets adding [44] using DEWÄKSS default configurations and similar preprocessing for all datasets. Data sets are filtered for expressed genes; genes which are expressed in fewer than 30 cells are removed from the data set. Any gene which does not have a minimum of 30 counts in at least one cell is also removed. For SAVER and DeepImpute, data is provided directly as integer counts; for all other methods, the expression data is log(x+1) transformed. All methods are used with default configurations. The hardware used is Memory: 31.1GiB, Processor: Intel Core i9-8950HK CPU @ 2.90GHz × 12, Graphics: GeForce GTX 1050 Ti with Max-Q Design/PCIe/SSE2, OS type: Ubuntu 18.04 64-bit, Swap disk: 64GB. If the method requires a parameter for the number of processors to use, it is set to 12, the maximum number of available cores.

## Supporting information

**S1 Section. Implementation of Molecular cross-validation [38].**
(PDF)

**S1 Table. Computational performance results for the tested methods.**
(PDF)

**S2 Table. All metrics computed for benchmarking deferentially expressed genes (DEGs), section 2.6.** The Gold Standard (GS) is knockout strains collected from the bulk deleteome data [39].
(PDF)

**S1 Fig. Heatmaps of normalized and denoised data on the RNAmix_CEL-seq2 dataset [29].** We use the R code provided in Tian et al. [43] to apply all normalization methods that we can successfully run on the RNAmix_CEL-seq2 dataset. We also apply our FTT-based preprocessing method on the data. We use the postprocessing and plotting scripts in [43] to generate plots for our analysis.
(PDF)

**S2 Fig. Denoising celltype annotated data from Hrvatin et al. [33].** The dataset contains 33 annotated celltypes in 48267 cells. a) Clustering performance metric computed against cell type clusters. Clusters were inferred with the Leiden algorithm [19]. The number at the bottom of each bar represents the inferred number of clusters. The Leiden algorithm's resolution parameter $r$ was set to 0.5 for MAGIC, 2 for pp and DeepImpute and 4 for DrImpute and SAVER to reduce or increase number of clusters, respectively, to comparative numbers given the previously annotated clusters. b) Silhouette score computed on 2 UMAP components using the 33 predefined cell type clusters. The input parameters to the umap algorithm is annotated with x and y labels of each panels row and column. c) Silhouette score computed on a varying numbers of PC components using the 33 predefined cell clusters. The number of PCs used corresponds to the number on the upper edge of the graph.
(PDF)

**S3 Fig. BM data [34].** Explained variance $\eta^2$ for each component $\Sigma$ (top row) and cumulative sum of $\eta^2$ for each component $\Sigma$ (bottom row). Each colored line indicates the data denoised with a specific set of parameters for the algorithms DEWÄKSS and MAGIC. X indicates the data without denoising. MAGIC truncates the number of possible components to the number of PCs used in the algorithm, which here equals 100. The right column only shows the first 100 components for the respective $\eta^2$ and $cumsum(\eta^2)$. MAGICd1 is removed from the top right-hand figure because it compresses the other lines. The explained variance is computed through the singular value decomposition and singular values lower than the numerical precision threshold are considered equal to 0 and removed. This threshold is determined by the criterion $\sigma_i \leq \sigma_1 \times \max(i, j) \times \epsilon$, where $i, j$ are the data dimensions and $\epsilon$ is the machine precision (numpy matrix_rank).
(PDF)

**S4 Fig. Epithelial-to-mesenchymal transition (EMT) data denoising.** A) The numbers of components needed to explain 99% and 90% of the variance in the data for different methods and hyperparameter values. MAGIC is run with 3 settings: default, t1 = one iteration, and "dewakss", using the optimal configuration found by DEWÄKSS. DEWÄKSS is run with 4 different settings: (i) optimal, as found by iterating over a range of hyperparameters (panel B and S6(B) Fig), (ii) oversmoothed, by running to $i = 4$ iterations, (iii) robust, i.e., using a different set of hyperparameters ($k = 100$, $PCs = 23$ selected as in S1 Section, $i = i_{\min MSE}$) and (iv) X base ($k = 100$, $i = i_{\min MSE}$), using normalized expression values instead of principal components as input to the kNN-G algorithm. B) The lowest MSE over all iteration values as a function of each DEWÄKSS parameter configuration, using connectivity graphs in the left plot and distances in the right plot. The lowest MSE configuration is found using distances with 100 PCs and $k = 100$ neighbors.
(PDF)

**S5 Fig. Optimal hyperparameter search for single-cell datasets.** The optimal number of PCs and neighbors is independent of the number of diffusion steps.
(PDF)

**S6 Fig. Optimal hyperparameter search for single-cell datasets.** Each figure shows the algorithm, using distances (top row) and connectivities (bottom row). Each panel is scaled to the maximum number of observed iterations (x-axis) for any configuration run, with the objective function value MSE on the y-axis. The colored lines indicate the number of PCs used as input in the kNN-G distance computation. Each column corresponds to the initial number of neighbors $k$ used for constructing the kNN-G.
(PDF)

**S7 Fig. Modified DEWÄKSS algorithm to not reset diagonal elements to 0 after each diffusion run.** This makes the objective function self referential and breaks the requirement setup by the noise2self principle. At the second diffusion step it becomes possible for cells to start influencing their own objective and therefore minimizes the objective function. This is akin to overfitting where each datapoint is best predicted by itself. This would encourage fewer neighbours as it will mirror the datapoint itself better. Note that diffusion turns out to be not needed and the methods converge for large enough k of initial neighbours.
(PDF)

**S8 Fig. Molecular cross validation (MCV) on Paul et al. [34].** MCV is here used to find optimal number of PCs to linearly project the data into. MCV with PCA shows that a linear projection on the data is best described by 13 components.
(PDF)

**S9 Fig. Differentially expressed genes (DEGs) between bulk and single cell data for deletion strains with few DEGs ($< 63$, 1% of yeast genome).** Top panel is delta AUROC between bulk DEGs and single cell DEGs ordered by adjusted p-value. Bottom panel is delta Jaccard index for single-cell DEGs at FDR = 0.01 for each separate deletion strain. *preprocessed* is count normalized and log-transformed with no denoising method. Delta is taken between denoised and preprocessed.
(PDF)

## Author Contributions

**Conceptualization:** Andreas Tjärnberg, Richard A. Bonneau.

**Data curation:** Andreas Tjärnberg, Christopher A. Jackson.

**Formal analysis:** Andreas Tjärnberg, Omar Mahmood, Christopher A. Jackson.

**Investigation:** Andreas Tjärnberg, Richard A. Bonneau.

**Methodology:** Andreas Tjärnberg.

**Project administration:** Andreas Tjärnberg.

**Resources:** Lionel A. Christiaen, Richard A. Bonneau.

**Software:** Andreas Tjärnberg, Omar Mahmood, Christopher A. Jackson, Giuseppe-Antonio Saldi.

**Supervision:** Andreas Tjärnberg, Kyunghyun Cho, Lionel A. Christiaen, Richard A. Bonneau.

**Validation:** Andreas Tjärnberg.

**Visualization:** Andreas Tjärnberg, Omar Mahmood, Christopher A. Jackson.

**Writing – original draft:** Andreas Tjärnberg, Omar Mahmood, Christopher A. Jackson, Kyunghyun Cho, Lionel A. Christiaen, Richard A. Bonneau.

**Writing – review & editing:** Andreas Tjärnberg, Omar Mahmood, Christopher A. Jackson, Kyunghyun Cho, Lionel A. Christiaen, Richard A. Bonneau.

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
