## [Decision Letter · Decision Letter 0]

25 Jun 2020

Dear Dr Tjärnberg,

Thank you very much for submitting your manuscript "Optimal tuning of weighted kNN- and diffusion-based methods for denoising single cell genomics data" for consideration at PLOS Computational Biology.

As with all papers reviewed by the journal, your manuscript was reviewed by members of the editorial board and by several independent reviewers. In light of the reviews (below this email), we would like to invite the resubmission of a significantly-revised version that takes into account the reviewers' comments.

We cannot make any decision about publication until we have seen the revised manuscript and your response to the reviewers' comments. Your revised manuscript is also likely to be sent to reviewers for further evaluation.

Sincerely,

Qing Nie

Associate Editor

PLOS Computational Biology

Weixiong Zhang

Deputy Editor

PLOS Computational Biology

Reviewer's Responses to Questions

**Comments to the Authors:**

Reviewer #1: This paper presents an interesting approach to denoising single-cell RNA-seq data based on (a) construction of a kNN graph (after PCA dimensionality reduction), (b) diffusion based denoising and (c) noise2self self-supervision procedure to select hyperparameters in this procedure: (i) dimension of PCA embedding, (ii) number of neighbors in the kNN graph, and (iii, presumably) number of diffusion steps in denoising.

The paper is well written, with appropriate benchmarking, analysis and interpretation. The method itself provides benefits in analysis due to the ability to provide a procedure for parameter selection based on self-supervision.

Nonetheless it would be helpful if the manuscript addressed some issues:

(a) comparison with noise2self pipeline for analysis. Some of the analysis of this paper mirror analyses from the noise2self paper (eg, the comparison of marker expression in optimal denoising vs. over smoothing). With that in mind, it would be helpful to see a comparison of the proposed algorithm with noise2self.

(b) the downstream analysis (variance retained) section is very limited. For instance, I don't follow how the authors use the number of PCs to recapture total variance after denoising as an argument for better performance. Please clarify in text.

(c) a more thorough analysis of differential expression can be performed beyond showing the effect on just a small number of marker genes. An analysis similar to the benchmark from Hou et al. https://www.biorxiv.org/content/10.1101/2020.01.29.925974v1 would be helpful.

(d) this is minor, but there is a statement that any kNN graph can be used but no experiment on the effect of the algorithm used to construct the kNN graph is provided. The UMAP method is fairly sophisticated and I wonder how much of the results presented here are based on how well that construction is made. Perhaps, a test with at least one other construction method would be helpful.

Reviewer #2: This is an overall well-structured paper proposing data imputation and denoising in single-cell transcriptomic data. The imputation is based on learning cell-cell similarity. It used the expression of a gene on neighboring cells of the cell of interest to impute the expression level of the same gene on this cell. The authors outlined their computational algorithm and use real scRNA-seq datasets to demonstrate its utility and compared with some existing imputation methods. My comments are as follows:

1. In the algorithm description on pages 10 and 11, Equations (2) and (5) have typo. Subscript k does not appear on the right side of both equations.

2. It is unclear the benefit of setting the diagonal elements of the Markov matrix M to 0. How does this step impact the performance in the real data analysis? Apparently, MAGIC does not require this step. The authors can compare their algorithm performance with and without this step.

3. There are many imputation methods available in literature. The authors compared the performance of their method to DrImpute and SAVER in the assessment of cell-cell correlation, and to DeepImpute and MAGIC in cell type clustering. In the last experiment to assess data variance, the authors compared to MAGIC. There is lack of comprehensive comparisons. I suggest to add a few more existing methods and compare them with the new one in every experiments outlined in this paper.

4. What is the computation cost? Please benchmark the computation time in comparison to other existing methods.

**Have all data underlying the figures and results presented in the manuscript been provided?**

Reviewer #1: Yes

Reviewer #2: Yes

PLOS authors have the option to publish the peer review history of their article (what does this mean?). If published, this will include your full peer review and any attached files.

Reviewer #1: No

Reviewer #2: No
---

## [Decision Letter · Decision Letter 1]

10 Nov 2020

Dear Dr Tjärnberg,

Thank you very much for submitting your manuscript "Optimal tuning of weighted kNN- and diffusion-based methods for denoising single cell genomics data" for consideration at PLOS Computational Biology. As with all papers reviewed by the journal, your manuscript was reviewed by members of the editorial board and by several independent reviewers. The reviewers appreciated the attention to an important topic. Based on the reviews, we are likely to accept this manuscript for publication, providing that you modify the manuscript according to the review recommendations.

Sincerely,

Qing Nie

Associate Editor

PLOS Computational Biology

Weixiong Zhang

Deputy Editor

PLOS Computational Biology

[LINK]

Reviewer's Responses to Questions

**Comments to the Authors:**

Reviewer #1: I appreciate the response from the authors and see all my questions addressed. While I recommend acceptance, better presentation of the new benchmark results would be great.

Reviewer #2: Most of my previous comments were addressed to my satisfaction except my first comment.

In Eq. (1), the matrix M represents cell-to-cell similarity and is a right-stochastic matrix with each row summing to 1. In the proposed new algorithm, the imputation was done by calculating the weighted average of express levels of the same gene in neighboring cells to recover the gene expression in a cell. So in Eq. (2), the imputed value of gene j in cell k is calculated by ∑_(k'=1)^z〖u_kk' x_k'j 〗, where the weight u_kk' is the kk'-th element of M and represents the similarity between cells k and k'. Additionally, the constrain on row sum of M was wrongly presented as column sum in the manuscript. Of course, if M is a square matrix, it would have the same constrains on row sum and column sum. Follow the similar argument, the subscripts in Eq. (5) also contained errors.

**Have all data underlying the figures and results presented in the manuscript been provided?**

Reviewer #1: Yes

Reviewer #2: Yes

PLOS authors have the option to publish the peer review history of their article (what does this mean?). If published, this will include your full peer review and any attached files.

Reviewer #1: No

Reviewer #2: No
---

## [Editor Report · Decision Letter 2]

28 Nov 2020

Dear Dr Tjärnberg,

We are pleased to inform you that your manuscript 'Optimal tuning of weighted kNN- and diffusion-based methods for denoising single cell genomics data' has been provisionally accepted for publication in PLOS Computational Biology.

Best regards,

Qing Nie

Associate Editor

PLOS Computational Biology

Weixiong Zhang

Deputy Editor

PLOS Computational Biology

---

## [Editor Report · Acceptance letter]

31 Dec 2020

PCOMPBIOL-D-20-00345R2 

Optimal tuning of weighted kNN- and diffusion-based methods for denoising single cell genomics data

Dear Dr Tjärnberg,

I am pleased to inform you that your manuscript has been formally accepted for publication in PLOS Computational Biology. Your manuscript is now with our production department and you will be notified of the publication date in due course.

With kind regards,

Jutka Oroszlan
